# A Single Dose of Beer after Moderate Aerobic Exercise Did Not Affect the Cardiorespiratory and Autonomic Recovery in Young Men and Women: A Crossover, Randomized and Controlled Trial

**DOI:** 10.3390/ijerph192013330

**Published:** 2022-10-16

**Authors:** Milana R. Drumond Santana, Yasmim M. de Moares Pontes, Cicero Jonas R. Benjamim, Guilherme da Silva Rodrigues, Gabriela A. Liberalino, Luana B. Mangueira, Maria E. Feitosa, Jessica Leal, Amanda Akimoto, David M. Garner, Vitor E. Valenti

**Affiliations:** 1Nucleus of Studies in Physiological and Pharmaceutical Sciences, School of Juazeiro do Norte, Juazeiro do Norte 63010-475, Brazil; 2Department of Internal Medicine, Ribeirão Preto Medical School, University of São Paulo, Ribeirão Preto 14049-900, Brazil; 3Autonomic Nervous System Center (CESNA), Sao Paulo State University (UNESP), Marilia 17525-900, Brazil; 4Cardiorespiratory Research Group, Department of Biological and Medical Sciences, Faculty of Health and Life Sciences, Oxford Brookes University, Headington Campus, Gipsy Lane, Oxford OX3 0BP, UK

**Keywords:** autonomic nervous system, beer, cardiovascular system, exercise, heart rate

## Abstract

**Background**: Beer is a choice beverage worldwide and is often consumed after sports for social interaction. Beer has been suggested for hydration after exercise, but the effects on cardiovascular and autonomic systems in men and women after effort are unknown. **Objectives**: We assessed the effect of beer absorption immediately after moderate exercise on heart rate (HR) variability (HRV) and cardiovascular recovery after effort in women and men separately. **Methods**: This is a crossover, randomized and controlled trial performed on 15 healthy female and 17 male adults submitted to two protocols on two randomized days: (1) Water (350 mL) and (2) Beer (350 mL). The subjects underwent 15 minutes seated at rest, followed by aerobic exercise on a treadmill (five minutes at 50–55% of maximum HR and 25 min 60–65% of maximum HR) and then remained 3 min stood on treadmill and 57 min seated for recovery from the exercise. Water or beer was consumed between four and ten minutes after exercise cessation. Blood pressure, HR and HRV were evaluated before exercise, during exercise and during recovery from exercise. **Results**: Systolic and diastolic blood pressure, HRV and HR changes during and following recovery from exercise were similar when women consumed beer or water. HR, systolic and diastolic blood pressures also returned to baseline levels in the beer and water protocols in males. Yet, parasympathetic indices of HRV recovery from exercise were comparable between protocols in males. **Conclusions**: Ingestion of 300 mL of beer did not significantly affect HRV and cardiovascular parameters following effort. Our data indicate that beer was safe for this population.

## 1. Introduction

Beer is a beverage habitually consumed after sports in recreational settings. This is since drinking a beer after a game is part of the social aspect of many sporting events [1,2,3,4]. Moreover, beer has been studied as an alternative for hydration after exercise and enhancement of daily energy and carbohydrate intake [5]. 

One of the important milestones in the area of alcohol and exercise research was the pioneering study by Burke et al., 2003 [6], who evaluated the effects of alcohol on muscle glycogen recovery after a prolonged exercise session in athletes. This study raised questions in the scientific literature that persist to the present day. In this work, the ingestion of 1.5 g/kg of alcohol did not interfere in the muscle glycogen resynthesis at 8 and 24 h after exercise, despite increasing serum triglycerides at 24 h after the exercise session. Smith et al. (2021) [7] evaluated the interference of alcohol on the use of muscle glycogen and found no differences in this outcome between vodka and water intake (control group). Based on the previously mentioned studies, the use and demand of energetic substrates during exercise do not seem to be affected by alcohol.

Alternatively, Dawson & Reid (1997) [8] had already demonstrated that there is a directly proportional relationship between alcohol consumption and fatigue, which culminates in a drop in performance. Still, alcohol consumption after physical exercise may decrease myofibrillar protein synthesis and, therefore, in the long term, may mitigate the anabolic effects of strength training (Parr et al., 2014 [9]). Regarding cardiovascular health, alcohol consumption can attribute an increase in the left atrium and consequent to an increase in the incidence of atrial fibrillation (McManus et al., 2016 [10]). It has been demonstrated that acute alcohol consumption is connected with the appearance of cardiac arrhythmias and sinus tachycardia (Brunner et al., 2017 [11]). Acute alcohol ingestion (1 g/kg body weight) caused decreased parasympathetic HRV indexes modulation and impaired the baroreceptor reflex sensitivity (Koskinen et al., 1994 [12]). Then, a previous study has found that in alcoholics, there are a significant decrease in linear time domain and HRV frequency indices, indicating that chronic alcohol consumption causes impairment of the autonomic modulation of cardiac activity (Kumar et al., 2014 [13]). These questionable effects of the relationship between alcohol and changes in cardiovascular behaviour suggest that beer consumption influences the autonomic control of heart rate (Julian et al., 2020 [14]).

However, the research literature has not fully considered the effect of beer on cardiovascular and autonomic recovery after exercise. This is hazardous, as the risk of sudden cardiovascular difficulties during and after exercise increases according to exercise intensity [15,16]. 

Thus, HR and HRV is a documented method that evaluates the oscillations between RR intervals. HRV provides information about the influence of the autonomic nervous system on heart rhythm regulation [17,18,19]. Healthy individuals display more prominent parasympathetic modulation and more significant variability between RR intervals. Throughout exercise, HRV decreases and returns quickly after exercise cessation [20]. In these cases, there is an improved transition capacity in the ANS response (vagal return) to the heart after stressful situations (e.g., physical exercise), causing a sudden fall in HR. Under this situation, autonomic recovery after exercise is a technique that evaluates cardiovascular risk following effort [17] and provides additional evidence concerning the risk of intervention before or after exercise [18]. Some environmental factors, such as high humidity in the same hot conditions (Abellán-Aynés et al., 2019 [21]) and high-intensity exercise slow HRV recovery after exercise (Kaikkonen et al., 2007 [22]). Women have a greater prominence of parasympathetic modulation on the heart in comparison to men, and female sex hormones (e.g., oestrogens) have a protective effect on cardiovascular health (ESHRE Capri Workshop Group, 2006 [23]). This physiological condition can affect autonomic reactivation responses after exercise and generate different results for men and women on the effects of a given intervention [24,25].

It is unclear in the scientific research literature if beer has an advantageous or negative effect on cardiovascular and autonomic parameters in exercise. In light of the above considerations, we highlight the following question: Is beer consumed after exercise able to modify the HRV recovery? Is the cardiovascular effect of beer on women equal to men? To answer this, we assessed the acute effects of beer intake directly after exercise on cardiovascular and HRV recovery of women and men. 

## 2. Materials and Methods

### 2.1. Trial Design

This is a transversal, crossover, randomized and controlled trial. The project was registered with the Brazilian Registry of Clinical Trials, which is accredited by the World Health Organization [26] (Protocol number: RBR-2j5294, http://www.ensaiosclinicos.gov.br/rg/RBR-2j5294, accessed on 8 July 2019). The study was performed at the Nucleus of Studies in Physiological and Pharmaceutical Sciences, School of Juazeiro do Norte, Juazeiro do Norte, CE, Brazil.

### 2.2. Participants

We primarily evaluated a total of 50 (25 men) healthy, physically active according to International Physical Activity Questionnaire (IPAQ), and who socially drink beer (one to two measures-maximum of 5 g alcohol, up to one day per week). We excluded subjects with resting HR > 100 bpm and body mass index (BMI) > 25 kg/m^2^, musculoskeletal, metabolic, cardiorespiratory and other related disorders under pharmacotherapies, and smokers [19]. We excluded women between the 10th and 15th days and between the 20th and 25th days of their menstrual cycle to avoid the influence of their luteal and follicular phases, respectively [27,28]. The final sample involved 32 subjects split into 17 men and 15 women to eliminate the influence of sex hormones on autonomic responses [29] (Figure 1).

### 2.3. Ethical Approval and Informed Consent

All experimental protocols were assessed and approved by the Research Ethics Committee in Research of the School of Juazeiro do Norte (Number 2.559.109). All participants signed a confidential informed letter of consent. All procedures were achieved in accordance with the 466/2012 resolution of the National Health Council of 12 December 2012. 

### 2.4. Initial Assessment

An anamnesis was commenced to confirm the absence of reported disorders and to evaluate the suitability of participating in the experimental protocol.

The individuals were inspected by obtaining age, height, mass, BMI, HR, systolic (SBP) and diastolic blood (DBP) pressures. The anthropometric measurements were logged according to Lohman et al. [30]. Stature was measured using a stadiometer (ES2020, Sanny, Brazil) with an accuracy of 0.1 cm. The mass was obtained via a digital scale (W200/5, Welmy, Brazil) with an accuracy of 0.1 kg. The BMI was computed by the mathematical formula: mass (kg)/height (m^2^). 

### 2.5. Outcomes

#### 2.5.1. Cardiovascular Variables

During the SBP and DBP measurements, the subjects remained seated. SBP and DBP were obtained indirectly by auscultation with a stethoscope (Premium, Barueri, SP, Brazil) and a calibrated aneroid sphygmomanometer (Premium, Barueri, SP, Brazil) on the subjects’ left arm [31]. The same investigator performed the SBP and DBP measurements. 

#### 2.5.2. HR and HRV Analysis

HRV analysis was achieved according to the Task Force of the European Society of Cardiology and the North American Society of Pacing and Electrophysiology guidelines [19]. The HR was accomplished beat-to-beat through an HR monitor (Polar RS800cx, Finland) with a sampling rate of 1 kHz. The RR intervals digital filtering was completed and complemented with manual filtering for the elimination of artefacts and we included only series with greater than 95% of sinus beats [32,33]. For data analysis, we designated stable series with 256 RR intervals [34,35].

HRV was scrutinized in the time domain through the standard deviation of normal interbeats (RR intervals) (SDNN) and the root mean square of successive differences (RMSSD). The frequency domain index was evaluated by the high frequency spectral component (HF) of the power spectral density (0.15 Hz to 0.4 Hz) in absolute units. We applied the Kubios HRV^®^ (Version 2.1) software package to compute these HRV indices [36].

### 2.6. Interventions

#### 2.6.1. Initial Assessment

The study was performed between 8:00 and 12:00 to standardize circadian effects [37] in a noiseless room with humidity between 40% and 70% and temperature between 22 °C and 28 °C. The subjects were instructed to refrain from drinking alcohol, caffeinated drinks or performing exhaustive exercise 24 h prior to each procedure. Subjects were advised to dress in suitable and comfortable clothing to permit the required physical effort and eat only a light meal two hours ahead of the procedures.

#### 2.6.2. Experimental Protocols

The experimental procedure interventions were completed over two days with an interval of 48 to 72 h between each session; permitting an acceptable recovery period for the subjects. 

As the initial step in the procedure, the subjects remained at rest, seated with spontaneous breathing for 15 min. Next, the subjects endured treadmill exercise with a slope of 1% in the first 5 minutes for warm-up (50–55% of maximal HR (HRmax): 220-age) [38]; after that, 25 minutes with increments of 0.5 km/h every minute until attainment of submaximal HR (60–65% of HRmax). Immediately after exercise, the subjects undertook three minutes standing on the treadmill and then seated for passive recovery for an additional 57 min, totalling 60 min of recovery [39]. The subjects ingested beer (300 mL, beer protocol) or water (300 mL, water protocol) in a typical glass between four and ten minutes after exercise. During recovery from exercise, the subjects remained seated silently with spontaneous breathing.

SBP and DBP were measured at the following times: Rest—15th minute of resting—and during recovery from exercise—Rec 15th minute, Rec 25th minute, Rec 35th minute, Rec 45th minute and, Rec 55th minute. HR were computed at the following times: Rest—10th minute of resting, during exercise (15th to 20th minute of exercise)—and during recovery from exercise—Rec 15th to 20th minute, Rec 25th to 30th minute, Rec 35th to 40th minute, Rec 45th to 50th minute and, Rec 55th to 60th minute [40,41].

HRV was recorded at the following times: Rest—10th to 15th minute of resting, during exercise (15th to 20th minute of exercise)—and during recovery from exercise—Rec 15th to 20th minute, Rec 25th to 30th minute, Rec 35th to 40th minute, Rec 45th to 50th minute and, Rec 55th to 60th minute [40,41] (Figure 2). 

### 2.7. Beer Composition

The beer was produced in Petropolis, RJ, Brazil and included the following ingredients and nutritional information: 350 mL, 138 Kcal (576 Kj), water, barley malt, undistorted cereals, hops, sodium isoascorbate (INS 316), sodium metabisulphite (INS 223), propylene glycol alginate (INS 405), alcohol: 4.5%, carbohydrates: 12.6 g, proteins: 0.24 g, fats: 0 g, cholesterol: 0 mg, sodium: 0 mg, potassium: 0 mg.

### 2.8. Sample Size

We applied the online software from the website www.lee.dante.br (accessed on 18 February 2020) to calculate sample size and we measured the RMSSD index. We considered a standard deviation of 12.8 ms and the magnitude of the difference was 14.11 ms, with alpha risk of 5% and beta risk of 80%. The sample size provided a minimum of 11 subjects per group.

### 2.9. Randomization

The subjects and the investigator were not blinded since the two interventions (beer and water) presented clear differences regarding colour and taste. The subjects were not informed about the sequence of the protocols. An investigator who contributed to the study made the random allocation sequence through random card selection, enrolled participants and assigned participants to their suitable interventions.

### 2.10. Statistical Methods

Data normality was assessed via the Shapiro–Wilk statistical test [42,43]. To compare cardiovascular variables and HRV, we computed repeated measures one-way analysis of variance (ANOVA1) followed by the Bonferroni post-test (parametric data) or Friedman followed by the Dunn’s post-test (non-parametric data). We performed statistical tests at the <0.01 (<1%) level of significance [44].

We calculated the effect size via Cohen’s d between two-time points. We assumed large effect size for values > 0.9, medium effect size for values between 0.9 and 0.5 and small effect size for values between 0.5 and 0.25 [45]. We enforced the Minitab software (Minitab, PA, USA) for the calculations.

## 3. Results

### 3.1. Sample Profile

The sample description regarding the age, mass, height and BMI of the subjects is described in Table 1. Table 1 illustrates the homogeneity for mass, age and BMI, except height which was significantly higher in men.

### 3.2. HR, SBP and DBP during Recovery from Exercise

We detected no significant difference between rest and exercise recovery about SBP and DBP for beer and water protocols in men. Regarding the water protocol, HR was increased during exercise vs. rest (Cohen’s d = 2.21, *p* < 0.001) and vs. 15–60 min after exercise (Rec 15–20 min: Cohens’ d = 2.07; Rec 25–30 min: Cohen’s d = 2.25; Rec 35–40 min: Cohen’s d = 2.21; Rec 45–50 min: Cohen’s d = 2.27; Rec 55–60 min: Cohen’s d = 2.33, *p* < 0.001) in men. In the beer protocol, HR was similarly increased during exercise vs. rest (Cohen’s d = 1.79, *p* < 0.001) and vs. 15–60 min during recovery from effort (Rec 15–20 min: Cohen’s d = 1.71; Rec 25–30 min: Cohen’s d = 1.80; Rec 35–40 min: Cohen’s d = 1.70; Rec 45–50 min: Cohen’s d = 1.78; Rec 55–60 min: Cohen’s d = 1.71, *p* < 0.001) in men (Figure 3).

In women, we detected no difference between rest and recovery from exercise for SBP and DBP. In the water protocol, HR was increased during exercise vs. rest (Cohen’s d = 2.87, *p* < 0.001) and vs. 15–60 min during exercise recovery (Rec 15–20 min: Cohen’s d = 3.20; Rec 25–30 min: Cohen’s d = 2.90; Rec 35–40 min: Cohen’s d = 3.08; Rec 45–50 min: Cohen’s d = 3.14; Rec 55–60 min: Cohen’s d = 2.65, *p* < 0.001) in women. In the beer protocol, HR was higher during exercise vs. rest (Cohen’s d = 3.52, *p* < 0.001) and vs. 15–60 min after effort (Rec 15–20 min: Cohen’s d = 3.23; Rec 25–30 min: Cohen’s d = 3.62; Rec 35–40 min: Cohen’s d = 3.21; Rec 45–50 min: Cohen’s d = 3.20; Rec 55–60 min: Cohen’s d = 3.21 *p* < 0.001) in women (Figure 4).

### 3.3. HRV during Exercise and following Effort

During the water protocol, SDNN was reduced during exercise vs. rest (Cohen’s d = 2.62, *p* < 0.001) and vs. 15–60 min following exercise end (Rec 15–20 min: Cohen’s d = 2.27; Rec 25–30 min: Cohen’s d = 2.84; Rec 35–40 min: Cohen’s d = 2.75; Rec 45–50 min: Cohen’s d = 3.20; Rec 55–60 min: Cohen’s d = 2.84, *p* < 0.001) in men. In the beer protocol, SDNN was lower vs. rest (Cohen’s d = 1.41, *p* < 0.001) vs. 15–20 min (Cohen’s d = 1.13) and vs. 55–60 min after exercise (Cohen’s d = 1.15, *p* < 0.001) in men (Figure 5).

We stated in the water protocol that RMSSD was decreased during exercise vs. rest (Cohen’s d = 2.21, *p* < 0.001) and vs. 15–60 min (Rec 15–20 min: Cohen’s d =1.71; Rec 25–30 min: Cohen’s d = 2.02; Rec 35–40 min: Cohen’s d = 2.23; Rec 45–50 min: Cohen’s d = 2.10; Rec 55–60 min: Cohen’s d = 2.35, *p* < 0.001) in men. In the beer protocol, RMSSD was lower during exercise vs. rest (Cohen’s d = 1.20, *p* < 0.001), vs. 35–60 min following exercise (Rec 35–40 min: Cohen’s d = 1.18; Rec 45–50 min: Cohen’s d = 1.34; Rec 55–60 min: Cohen’s d = 0.69, *p* < 0.001) in men (Figure 5).

For HF (ms^2^) in the water protocol, it decreased during exercise vs. 15–20 min (Cohen’s d = 1.29, *p* < 0.001), vs. 35–60 min after exercise (Rec 35–40 min: Cohen’s d = 1.61; Rec 45–50 min: Cohen’s d = 1.61; Rec 55–60 min: Cohen’s d = 1.27, *p* < 0.001) in men. The same index in the beer protocol was reduced during exercise vs. 55–60 min following exercise cessation (Cohen’s d = 0.66, *p* < 0.001) (Figure 5).

In women, the water protocol SDNN decreased during exercise vs. rest (Cohen’s d = 2.47, *p* < 0.001) and vs. 15–60 min after effort (Rec 15–20 min: Cohen’s d = 2.63; Rec 25–30 min: Cohen’s d = 2.11; Rec 35–40 min: Cohen’s d = 2.89; Rec 45–50 min: Cohen’s d = 2.65; Rec 55–60 min: Cohen’s d = 2.52, *p* < 0.001). In the beer protocol, SDNN lessened during exercise vs. rest (Cohen’s d = 2.59, *p* < 0.001) and vs. 15–60 min following exercise cessation (Rec 15–20 min: Cohen’s d = 1.99; Rec 25–30 min: Cohen’s d = 2.10; Rec 35–40 min: Cohen’s d = 1.85; Rec 45–50 min: Cohen’s d = 2.17; Rec 55–60 min: Cohen’s d = 2.25, *p* < 0.001) (Figure 6) in women.

Concerning RMSSD in the water protocol, it declined during exercise vs. rest (Cohen’s d = 2.40, *p* < 0.001) and vs. 15–60 min after effort (Rec 15–20 min: Cohen’s d = 3.12; Rec 25–30 min: Cohen’s d = 2.31; Rec 35–40 min: Cohen’s d = 2.50; Rec 45–50 min: Cohen’s d = 2.73; Rec 55–60 min: Cohen’s d = 2.31, *p* < 0.001) in women. In the beer protocol, RMSSD reduced during exercise vs. rest (Cohen’s d = 2.71, *p* < 0.001) and vs. 15–60 min after exercise end (Rec 15–20 min: Cohen’s d = 2.32; Rec 25–30 min: Cohen’s d = 2.49; Rec 35–40 min: Cohen’s d = 2.02; Rec 45–50 min: Cohen’s d = 2.53; Rec 55–60 min: Cohen’s d = 2.21, *p* < 0.001) in women (Figure 6).

Relating to HF (ms^2^) in the water protocol, it lessened during exercise vs. 35–40 min (Cohen’s d = 1.09, *p* < 0.001) and vs. 55–60 min following effort (Cohen’s d = 1.74, *p* < 0.001) in women. In the beer protocol, HF (ms^2^) reduced during exercise vs. rest (Cohen’s d = 1.54, *p* < 0.001), vs. 35–40 min (Cohen’s d = 1.08, *p* < 0.001) and vs. 55–60 min after exercise cessation (Cohen’s d = 1.49, *p* < 0.001) (Figure 6).

## 4. Discussion

This study was commenced to evaluate the acute effects of beer intake immediately after exercise on cardiovascular and HRV recovery. We revealed that: (1) beer ingestion following effort did not affect cardiovascular and HRV recovery in women and men; (2) SBP, DBP and HR returned to baseline after exercise and were unaffected by beer consumption in men; (3) The ingestion of up to 350 mL beer ingestion following exercise seems safe for this specific population.

Preceding studies have investigated the influence of beer on physiological parameters in exercise. Castro-Sepulveda et al. [46] stated that alcoholic beer before exercise negatively affected sports performance and health as it reduced plasma Na+ and increased plasma K+ during exercise. An additional study evaluated the relationship between beer consumption and resting HRV in women survivors of acute myocardial infarction and those who underwent a revascularization procedure [47]. HR was logged for 24 h and the investigators did not observe an important relationship between resting HRV and beer consumption. It was unclear thus far about the acute impact of beer after exercise on HRV recovery, which is a typical situation for social interaction after recreational sports.

Consistent with our results, no significant change was observed between resting vs. exercise recovery for SBP and DBP when men and women consumed beer or water. This is probably because the subjects performed an exercise intensity (60–65% HRmax) that did not attain the anaerobic threshold [48]. 

We demonstrated that HR increased during exercise in the beer and water protocols. When men and women consumed water, HR recovered to the baseline between 15 to 20 min after exercise cessation. Likewise, when they ingested beer, HR returned to resting values after between 15 to 20 min of effort. In this way, we propose that beer did not influence HR during recovery from exercise.

For HRV in women, we reported that SDNN and RMSSD indices recovered to baseline levels between 15 and 20 min following exercise when they consumed water or beer. HF band returned to resting values between 35 and 40 min after effort in the water and beer protocols. Based on these results, we suggest that beer did not impact HRV recovery from aerobic moderate exercise in women.

When men were submitted to aerobic exercise, the SDNN index returned to baseline values between 15 and 20 min after exercise in the water protocol. When they consumed beer, the SDNN index was totally recovered to resting levels between 55 and 60 min after exercise cessation. We also detected that the RMSSD and HF band recovered to resting values between 15 and 20 min following exercise when men consumed water. When men ingested beer, the RMSSD returned to baseline between 35 and 40 min and the HF band returned to resting values between 55 and 60 min after exercise. The cited results indicate that beer intake after exercise did not significantly affect HRV recovery in men. This is because HRV was not stable 60 min following exercise in the beer and water protocols.

In this line, a review conducted by Koob et al. [49] reinforced that alcohol ingestion over the short term increases the hypothalamic-pituitary adrenal activity. The hypothalamic paraventricular nucleus is an appropriate area for the integration of sympathetic outflow [50]; higher activity of this region modulates excitatory cardiovascular responses [51]. Moreover, Brunner et al. [11] completed an observational, cross-sectional cohort study to evaluate the connection between alcohol intake and cardiac arrhythmia. The study revealed an important connection between acute alcohol consumption and sinus tachycardia. 

With this in mind, the effect of alcoholic and non-alcoholic energy drinks on post-exercise HR recovery and HRV has been formerly studied [52]. The investigation was conducted on 10 healthy volunteers (five men) who submitted to a maximal bicycle ergometer exercise planned to last for up to 15 min. The subjects consumed 0.75 L energy drink or 0.75 L energy drink mixed with vodka (0.4 g of ethanol/kg) and exercised 30 min later. Whilst no significant clinical indication of arrhythmia was reported, the study demonstrated that energy drinks mixed with alcohol delayed HRV and HR recovery. 

We previously hypothesized that women would present different HRV responses induced by beer. Our hypothesis was based on the cardioprotective mechanisms of female sexual hormones [53,54]. Prior studies emphasized the potential impact of natural oestradiol metabolites on left ventricular hypertrophy [54] and the influence of oestrogen on obese-insulin resistance [53]. Nevertheless, this hypothesis was not supported by our data.

There are studies indicating that beer has alternate effects in men and women. Recently, Camps et al. [55] assessed the influence of acute beer consumption on neural activation. The investigation suggested differences between genders regarding cerebral blood flow after beer and soda drink digestion, with more intense responses in men [56]. 

Some relevant issues are noteworthy. We were unable to complete a double-blind or single-blind placebo design as the volunteers could differentiate water or beer content because of the drinks’ taste. The specific components of beer (alcohol, barley malt, cereals, hops) were mixed as this is a new study focused on alcoholic beer. We encourage further studies to evaluate the impact of non-alcoholic beer on HRV recovery after exercise. Another limitation of this study was the additives present in the beer composition, such as “sodium isoascorbate (INS 316)”, “sodium metabisulphite (INS 223)”, “propylene glycol alginate (INS 405)”, that can be interference with the variables analysed in the study. Alternatively, to our knowledge, there is no interaction between these additives to cardiovascular and autonomic evaluated in our research. 

The submaximal aerobic exercise was chosen because it is the most prevalent intensity in which the global population does attain exercise (e.g., walking). However, this can also be considered a limitation because data cannot be extrapolated to exercise with vigorous-intensity (e.g., powerlifting, soccer). We recommend further studies to simulate specific sports to achieve accurate results from beer in cardiovascular recovery after effort.

We did not assess urine output, which would provide important information regarding the participants’ levels of hydration. Yet, no subject needed to urinate and maintained an empty bladder from the outset. We decided not to evaluate the low frequency (LF) or the LF/HF ratio as they have been revealed to be fundamentally flawed and empirically unsupported to represent the sympatho-vagal balance [57,58]. We chose to estimate the effect of beer after exercise as this is the most usual time that sporting players drink it for social interaction.

Our study’s strong points include (1) a randomized controlled design; (2) an extra detailed analysis of HRV recovery (every five minutes following beer consumption), and; (3) a more demanding *p*-value for significant differences (*p* < 0.01, <1%) that improves reproducibility and facilitates replication of our data [44]. The inability to blind participants between interventions is one of the limitations of this study.

## 5. Conclusions

Our results indicate that 300 mL of beer consumption immediately after exercise did not affect HRV and cardiovascular recovery following effort. In this scenario, the data suggest that a single dose of beer after exercise is safe for young women and men adults.

## Figures and Tables

**Figure 1 ijerph-19-13330-f001:**
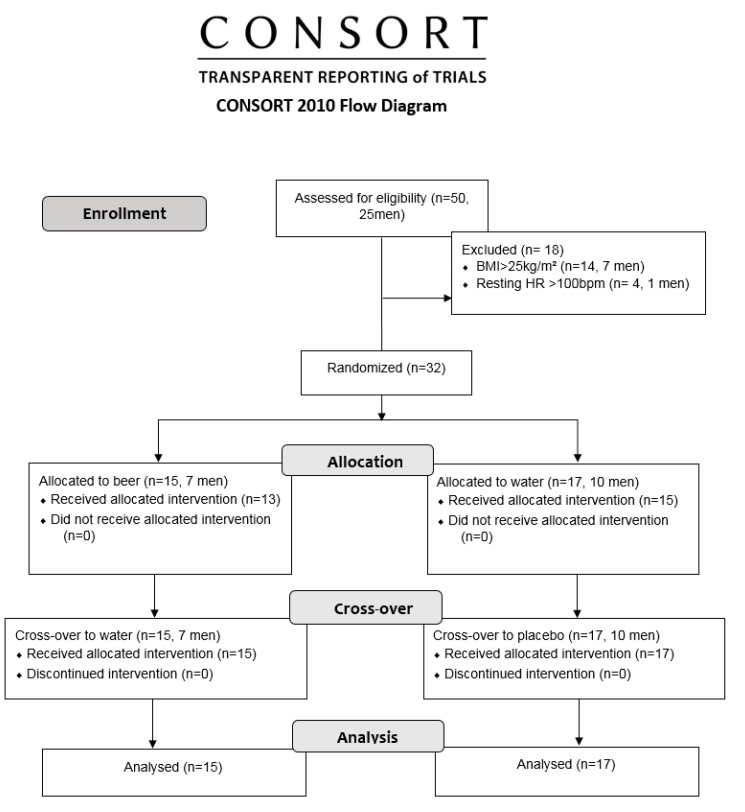
Flowchart representation of the study phases (enrolment, allocation, crossover and analysis).

**Figure 2 ijerph-19-13330-f002:**
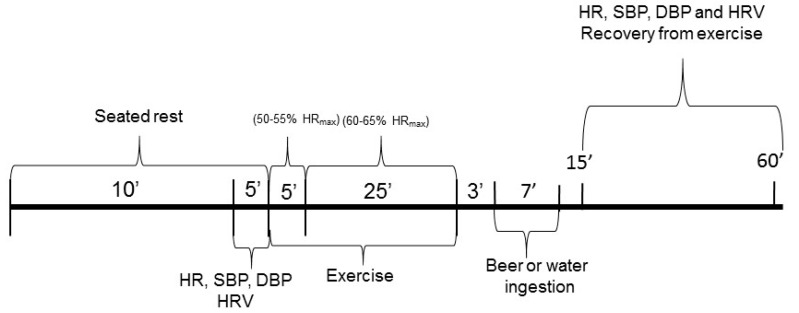
Study Design.

**Figure 3 ijerph-19-13330-f003:**
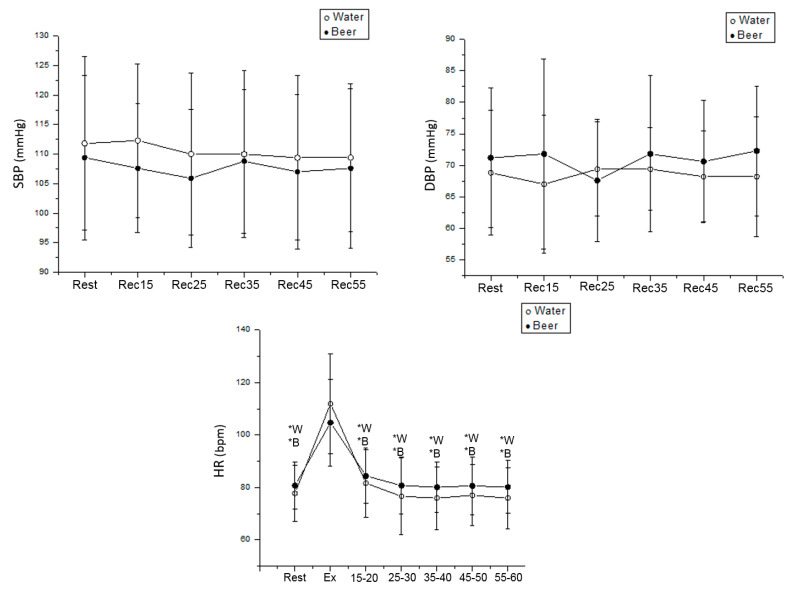
HR, SBP and DBP values were obtained at rest and during recovery from moderate aerobic exercise protocol in men. *W: Values with significant differences in relation to exercise (*p* < 0.01) for water protocol; *B: Values with significant differences in relation to exercise (*p* < 0.01) for beer protocol.

**Figure 4 ijerph-19-13330-f004:**
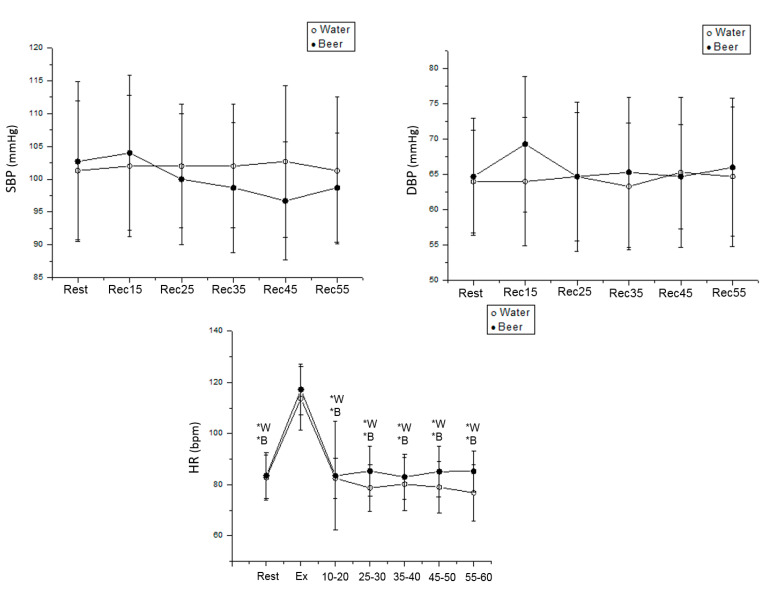
HR, SBP and DBP values were obtained at rest and during recovery from moderate aerobic exercise protocol in women. *W: Values with significant differences in relation to exercise (*p* < 0.01) for water protocol; *B: Values with significant differences in relation to exercise (*p* < 0.01) for beer protocol.

**Figure 5 ijerph-19-13330-f005:**
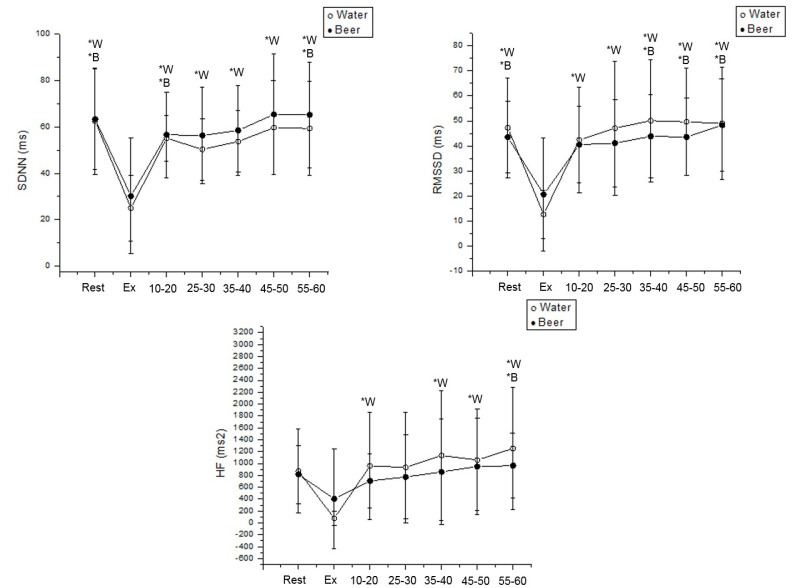
SDNN, RMSSD and HF values were obtained at rest and during recovery from moderate aerobic exercise protocol in men. *W: Values with significant differences in relation to exercise (*p* < 0.01) for water protocol; *B: Values with significant differences in relation to exercise (*p* < 0.01) for beer protocol.

**Figure 6 ijerph-19-13330-f006:**
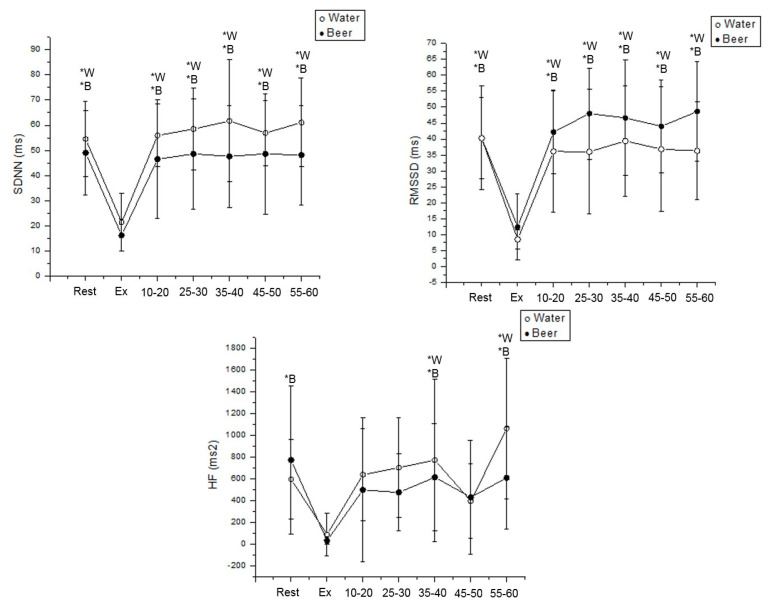
SDNN, RMSSD and HF values were obtained at rest, during exercise and during recovery from moderate aerobic exercise protocol in women. *W: Values with significant differences in relation to exercise (*p* < 0.01) for water protocol; *B: Values with significant differences in relation to exercise (*p* < 0.01) for beer protocol.

**Table 1 ijerph-19-13330-t001:** Characterization of the sample regarding age, mass, height and BMI. Mean ± standard deviation [minimum–maximum].

Variables	Men	Women	*p*-Value	Cohen’s d
**Mass (kg)**	65.22 ± 7.65 [46.1–78.6]	60.99 ± 8 [48.9–79]	0.13	-
**Height (m)**	1.72 ± 0.06 [1.63–1.82]	1.62 ± 0.06 [1.56–1.79]	<0.0001	1.67
**Age (years)**	20.94 ± 2.33 [18–26]	22.67 ± 3.56 [19–32]	0.13	-
**BMI (kg/m^2^)**	21.92 ± 2.68 [17.35–24.8]	23.12 ± 1.95 [18.86–24.9]	0.16	-

**Legend:** BMI: body mass index; kg: kilogram; m: meters.

## Data Availability

Not applicable.

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
