# Peer review of "A Single Dose of Beer after Moderate Aerobic Exercise Did Not Affect the Cardiorespiratory and Autonomic Recovery in Young Men and Women: A Crossover, Randomized and Controlled Trial"

_ijerph, 2022, doi:10.3390/ijerph192013330_

Round 1

Reviewer 1 Report

Line 28. What cardiorrespiratory paremeters were assessed?

Line 28. The abbreviation HRV has been previously described, don't describe it again here. However, SDNN, RMSSD and HF were not previously described and must be described before using the abbreviations. Why are you indicating units for HF and not for SDNN and RMSSD? 

Line 33. Indicate that the consumption is 300mL in the conclusions. The fact of beer ingestion can be misunderstood if quantities are not indicated.

Line 49. HR and HRV have been described previously.

Line 41. Is this all the literature about the use of beer in exercise?

Line 52. Is this fact right? Author must cite previous studies that have observed slower recoveries of HRV after exercise and discuss this idea in this paragraph, mentioning what factors can influence on the HRV recovery

Abellán-Aynés, O., López-Plaza, D., Alacid, F., Naranjo-Orellana, J., & Manonelles, P. (2019). Recovery of heart rate variability after exercise under hot conditions: the effect of relative humidity. Wilderness & Environmental Medicine30(3), 260-267.

Kaikkonen, P., Nummela, A., & Rusko, H. (2007). Heart rate variability dynamics during early recovery after different endurance exercises. European journal of applied physiology102(1), 79-86.

The introduction is poor, authors must deepen on the topic related to beer intake and its effects on physiological responses during and after exercise. Furthermore, if HRV is being the main characteristic assessed in the study, authors must better describe HRV in the introduction.

Line 106. What is IBIs? I didn't see it before in the text

Line 132-139. I think it would be less confusing if you use the terms Rec20 to indicate recovery between 15th and 20th minute rather than rec1 (this would seem recovery after 1 minute). Make this change for all the recovery moments in each variable.

Line 139. Bring figure 2 closer to this point.

In the results section, authors are presenting the results spread by gender. Thus, along the whole paper it should have been described the differences between men and women in the recovery after exercise. Those, the title, abstract and introduction must point out that the study makes such distintion between gender and the introduction must deepen on the differences between gender on recovery after exercise.

Line 173-220. In the results section you are just mentioning the effect size, which is relevant. However, if you performed the ANOVa analysis with the Bonferroni Pos-Hoc you must also present the p-values for each comparison. Also remember to show rec20, rec30 rather than rec1, 2...

Line 221. Did you mean "Mean ± standard deviation"?

Reviewer 2 Report

An interesting comparison of the HRV, HR and BP effects of beer vs water post exercise.  Nice to see this type of investigation about supposed safe alcohol usage post exercise.

Abstract - capitalize "we" line 20

HR methodology - what version of Kubios, what artifact correction method was used and what was the acceptable artifact percentage to be included in the study.

Comments should be made on the HRV data in relation to exercise intensity in the discussion.  SDNN has been used as a method of detecting the VT1 (doi: 10.1055/s-2007-989423).  Values presented here seem consistent with intensity below the VT1 which is relatively mild.  This should be commented on as a limitation.  Further study could be done as to the effects of alcohol consumption post higher intensities typically seen with semi competitive events (1/2 marathon, recreational cycling, soccer matches etc) - which possibly could lead to an even more profound differentials.

Effects of “sodium isoascorbate (INS 316), sodium metabisulphite (INS 223), propylene glycol alginate (INS 405)”, should also be considered in the limitations.  This was not a direct water vs alcohol comparison.

Round 2

Reviewer 1 Report

Line 29. You are still defining the meaning of HRV in the abstract twice. Delete de last one.

Line 58. The sentence is wrongly written, "It HAS been demonstrated that..."

Line 60. Wrongly written, "a previous study HAS...."

Line 61. It is interesting mentioning the long-term effect of alcohol (i.e. alcoholics) on HRV. However, this study focuses on acute effects, so more introduction to acute effects of alcohol intake must be mentioned in the text.

You wrote the same paragraph twice "Women have a greater prominence of parasympathetic modulation on the heart compared to men, and female sex hormones (e.g. estrogens) have a protective effect on cardiovascular health. This physiological condition can affect autonomic reactivation responses and generate different results for men and women on the effects of a given intervention"

In line 74 authors still mention that the recovery of HRV after exercise happens quickly. Authors must debate in the introduction this fact. They must cite these two studies in order to explain what factors such as intensity and external environment can affect the HRV recovery.

Abellán-Aynés, O., López-Plaza, D., Alacid, F., Naranjo-Orellana, J., & Manonelles, P. (2019). Recovery of heart rate variability after exercise under hot conditions: the effect of relative humidity. Wilderness & Environmental Medicine, 30(3), 260-267.

Kaikkonen, P., Nummela, A., & Rusko, H. (2007). Heart rate variability dynamics during early recovery after different endurance exercises. European journal of applied physiology, 102(1), 79-86.

Line 74. There is a cite "11" with a number in superscript. This is not consistent with the journal's format. This issue is present in many occasions along with the manuscript with the new changes.

Generally, in the introduction, still, there is a lack of explanation of the concept of HRV and the acute effects of alcohol intake on other physiological variables related to exercise.

Line 99. Cite in number and superscript.

The title of figure 1 is poor. Only "Flowchart" can't be the title of a figure. Define what the flowchart represents
